# A Study on the Application of Discrete Wavelet Decomposition for Fault Diagnosis on a Ship Oil Purifier

Songho Lee [1,2,†], Taehyun Lee [1,†], Jeongyeong Kim [1], Jongjik Lee [1], Kyungha Ryu [1], Yongjin Kim [1,*] and Jong-Won Park [1,*]

1 Department of Reliability Assessment, Korea Institute of Machinery and Materials, Daejeon 34103, Korea; dlrm741@kimm.re.kr (S.L.); thlee07@kimm.re.kr (T.L.); jykim8792@kimm.re.kr (J.K.); ljjik@kimm.re.kr (J.L.); khryu@kimm.re.kr (K.R.)
2 School of Mechanical Engineering, Pusan National University, Busan 46241, Korea
* Correspondence: yjkim2014@kimm.re.kr (Y.K.); jwpark@kimm.re.kr (J.-W.P.)
† These authors contributed equally to this work.

**Abstract:** With the development of the Internet of things, big data, and AI leading the 4th industrial revolution, it has become possible to acquire, manage, and analyze vast and diverse condition signals from various industrial machinery facilities. In addition, it has been revealed that various and large amounts of signals acquired from the facilities can be utilized for fault diagnosis. Currently, while data-driven fault diagnosis techniques applicable to the facilities are being developed, it has been tried to apply the techniques for the development of fully autonomous ships in the shipbuilding and shipping industry. Since the autonomous ships must be able to detect and diagnose the failures on their own in real time, the overall research is required on how to acquire signals from the ship facilities and use them to diagnose their failures. In this study, a fault diagnosis framework was proposed for condition-based maintenance (CBM) of ship oil purifiers, which are an auxiliary facility in the engine system of a ship. First, an oil purifier test-bed for simulating faults was built to obtain data on the state of the equipment. After extracting features using discrete wavelet decomposition from the data, the features were visualized by using t-distributed stochastic neighbor embedding, and were used to train support vector machine-based diagnostic models. Finally, the trained models were evaluated with *Accuracy* and $F_1$ *score*, and some models scored 0.99 or higher, confirming high diagnostic performance. This study can be used as a reference for establishing CBM system and fault diagnosis system. Furthermore, this study is expected to improve the safety and reliability of oil purifiers in Degree 4 MASS.

**Keywords:** fault diagnosis; discrete wavelet transform (DWT); wavelet packet transform (WPT); condition-based maintenance (CBM); oil purifier

## 1. Introduction

Recently, with the development of the Internet of things, big data, and artificial intelligence (AI), which lead the 4th industrial revolution, the development of Maritime Autonomous Surface Ships (MASS) has become an issue in the shipbuilding and shipping industry. MASS is a vessel that has no or minimized human intervention on the surface of the water, which has a degree of 1 to 4 according to automation level [1]. Currently, many studies have been conducted to develop the Degree 4 MASS, that is, a fully autonomous ship. Since the Degree 4 MASS operates with no or minimal human intervention, it should be able to recognize unexpected events in the ship engine system and take action by itself. Therefore, it is necessary to develop self-diagnosis techniques [2,3].

One of the factors that should be considered in the development of self-diagnostic techniques is prognostics and health management (PHM). In the past, the ships were maintained by fault alarm-based breakdown maintenance (BM) or time-based maintenance (TBM), referring to the past maintenance history [4]. There are several disadvantages

of relying on early replacement (resource cost) or addressing accidental failures (time and resource cost). By using condition-based maintenance combined with PHM, we can overcome these disadvantages. It monitors complex system in real time, detects abnormalities in the system in advance, and predicts future failures beforehand. Through these processes, it is possible to improve safety and reliability of facilities, and to establish efficient maintenance policies [5].

Condition monitoring with PHM begins with establishing state database of the facilities. After pre-processing the database, it proceeds in the order of diagnosing the current state and predicting the future failure [6]. In this process, factors affecting successful diagnosis and prognosis are data, models, and algorithms [7]. Especially, acquiring and preprocessing data are important. Although the other processes are well organized, the diagnosis/prognosis performance can be poor due to the poor database [8]. However, preparing good quality database are still challenging because the data are scarce and unbalanced. [9]. Also, many types of sensors provide various signals about the normal condition of a facility, but data on failed conditions are limited or not provided. It is because most facilities have been employed using conservative maintenance policies such as BM or TBM, due to the costs and the catastrophic results which can emerged [2,3].

One strategy to address this is to use a physical model to derive reliable results using a small amount of data. This is a hybrid approach which uses data and physics, such as physics-informed neural network (PINN) [10,11]. Another strategy is to acquire the data directly from the faulty system, using a testing device for simulation rather than the real one. Although it is expensive to build a test-bed, which is a realistic imitation system to simulate failure, it is possible to efficiently develop a framework for fault diagnosis and prognosis because failures are simulated and data are acquired on the test-bed. When PHM needs to be introduced in a system with high complexity and high cost due to failures, it can be efficient to acquire reliable data from the test-bed for the development of a fault diagnosis and prognosis framework.

An oil purifier is one of auxiliary equipment of a ship engine system, and it separates crudes oil, including fuel oil and lubricating oil, into purified oil and sludge. Failure of the oil purifier is highly fatal, because when it has a problem with the outlet flow and cleaning condition of the outlet oil, it may lead to failure of the entire ship as well as other engine system facilities [2,12]. The failure of the oil purifier often starts with a faulty component [3], but it is hard to detect the fault of the component inside the equipment. Despite the limitation of BM and TBM, for the above reasons, they have been implemented so far. However, for the development of Degree 4 MASS, it is essential to develop CBM with PHM for oil purifiers in ships from now on.

In this study, we conducted research in two stages with the purpose of developing a fault diagnosis framework based on condition monitoring of marine oil purifiers. In the first stage, the normal and faulty state data of the oil purifier were acquired. We built a test-bed for land-based simulation of the oil purifier, and acquired condition data by simulating faults based on fault analysis. In the second stage, the fault diagnosis framework for oil purifiers was presented. First, the signal was preprocessed using discrete wavelet decomposition, and several wavelet spectrum measures were calculated. After learning the support vector machine-based architecture, the diagnosis performance for each learning method was evaluated with a few evaluation metrics. Through the stages, the article provides not only a fault diagnosis framework for the equipment with complex systems, but also new perspective on how to acquire and handle signals for application of CBM using the framework.

So, the main contributions of this article can be listed as follows:

- Proposed the measuring instrument setup and DAQ method to apply a CBM policy to oil purifiers in real ships;
- Established the condition database of the oil purifier based on fault simulation using the realistic land-based test-bed;

- Proposed the fault diagnosis framework using discrete wavelet decomposition and support vector machine.

## 2. Methodology for Fault Diagnosis

The purpose of this study is to diagnose the state of the oil purifier based on condition monitoring. To achieve this, we adopted a data-driven approach that infers the system status from the collected data. In this section, discrete wavelet decomposition and support vector machine for multiclass classification is introduced.

### 2.1. Discrete Wavelet Decomposition

Methods to analyze waveform-type data such as vibration can be divided into time domain analysis, frequency domain analysis, and time-frequency domain analysis [7]. Among them, time-frequency domain analysis has advantages of being able to analyze non-stationary signals because the frequency analysis is performed and repeated in a certain time window within the original signal. Short-time Fourier transform (STFT) and wavelet transform (WT) are representative of time-frequency domain analysis [12]. STFT is a method in which a signal in the time domain is cut into a certain time length, and then fast Fourier transform (FFT) is performed for each time section. However, since the original signal is cut with a time window of a certain length, there is a trade-off problem between time resolution and frequency resolution.

Wavelet transform (WT) is a method that compensates for the shortcomings of STFT. As scale and translation parameters change, two orthonormal basis functions, the scaling function and the wavelet function, change, and the original signal is decomposed into sub-signals with various resolution conditions. Therefore, multi-resolution analysis (MRA) with improved time resolution and frequency resolution is possible [13,14]. WT is usually categorized as continuous wavelet transform (CWT), discrete wavelet transform (DWT), and wavelet packet transform (WPT) [15,16]. In CWT, the scale and translation parameters of wavelet and scaling functions change continuously. Although detailed MRA is possible through CWT, there are disadvantages in that redundant information is generated and computational cost is increased. A way to solve this problem is to use DWT, especially to binarize the parameters. It can reduce the computational cost and, at the same time, avoids the loss of information contained in the signal [15,17]. In DWT, the original signal is decomposed through high pass filter (HPF) and low pass filter (LPF) and then down-sampled in half, as shown in Figure 1. Detail and approximation coefficients can be obtained through this process. After that, DWT process is repeated using the obtained approximation coefficients as an input [18,19].

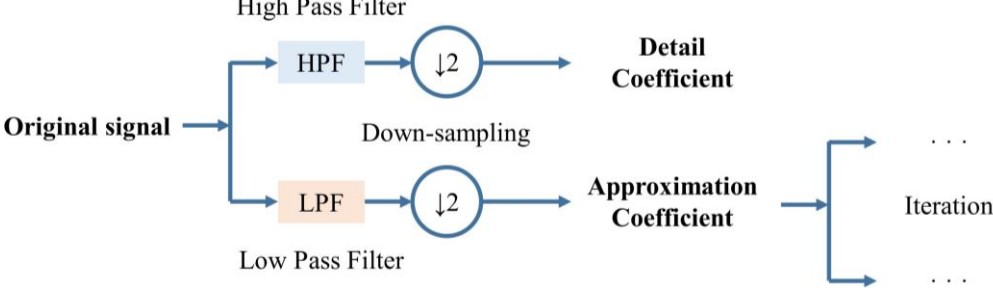

**Figure 1.** Schematic diagram describing the mechanism of DWT.

Similar to DWT, WPT decomposes a signal into a low-frequency component and a high-frequency component, but it is the technique that continuously decomposes not only the approximation coefficients but also the detail coefficients that have passed the HPF. Therefore, it can be called a generalized discrete wavelet decomposition compared to DWT. Figure 2 shows maximum decomposition level $J = 3$ of WPT, which is called wavelet decomposition tree. As shown in Figure 1, the original signal is filtered through HPF

and LPF in the first decomposition $j = 1$, and then down-sampled and converted into detail coefficient $D_1$ and approximation coefficient $A_1$. After that, in WPT, not only the approximation coefficients but also the detail coefficients are decomposed up to the target decomposition level $J$. Through repetition of this process, the original signal is decomposed into nodes having $2^J$ sub-bands, and it can be confirmed that each node has a frequency band of the same size [18,20]. The coefficients shown in red in Figure 2 mean the coefficients that can be obtained through DWT.

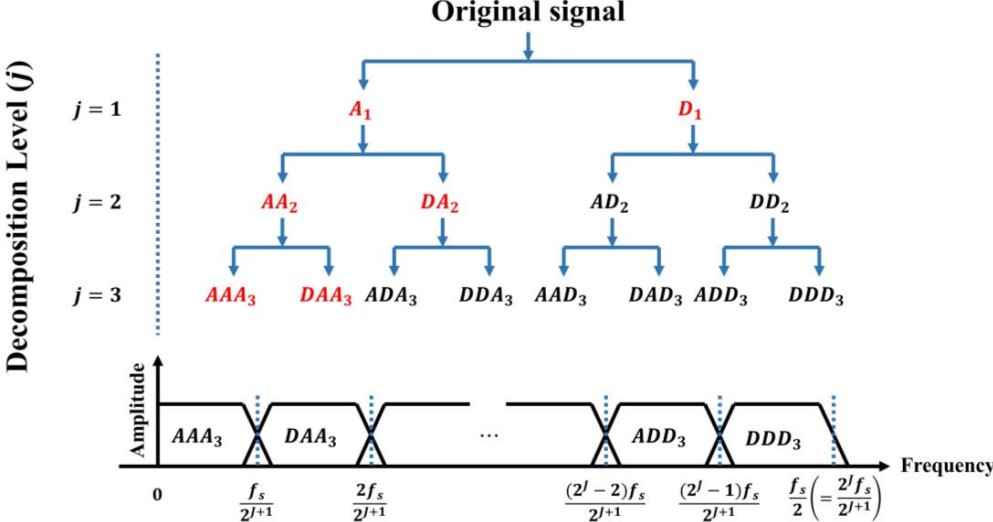

**Figure 2.** Schematic diagram describing the wavelet decomposition tree and its frequency band.

After decomposing the signal using DWT and WPT, wavelet spectrum can be used to extract features from the decomposed signal. Wavelet spectrum is a kind of energy expression form designed for calculation of Hurst parameter [13]. Through the method, we can estimate $\log_2 \left( \frac{1}{n_j} \sum_{k=1}^{n_j} d_{j,k}^2 \right)$, called the wavelet spectrum, by using the sample energy of the wavelet coefficients $d_{j,k}$, assuming $d_{j,k}$ is the $k$th detail coefficient the decomposition level $j$ ($j = 1, \cdots, J$, $k = 1, \cdots, n_j$) [14]. In addition, to identify additional characteristics of the energy from statistical point of view, other wavelet spectrums suggested in previous studies can be calculated, which are shown in Table 1 [21]. The mean measure is the binary logarithm of the mean energy of the sampled wavelet coefficients. Median measure has the advantage of being relatively robust to noise compared to mean measure. Additionally, the variation of the wavelet coefficients can be considered by using variance and interquartile range (IQR) measure.

**Table 1.** Wavelet spectrum based on discrete wavelet transform.

| Measure | Wavelet Spectrum |
|---|---|
| Mean | $w_j = \log_2 \left[ \frac{1}{n_j} \sum\limits_{k=1}^{n_j} d_{j,k}^2 \right]$ |
| Median | $w_j = \mathrm{median} \left( d_{j,k}^2 \right)$ |
| Variance | $w_j = \log_2 \left[ \frac{1}{n_j-1} \sum\limits_{k=1}^{n_j} \left( d_{j,k}^2 - \bar{d}_{j,k}^2 \right)^2 \right]$ |
| Interquartile range (IQR) | $w_j = Q_3 \left( d_{j,k}^2 \right) - Q_1 \left( d_{j,k}^2 \right)$ |

### 2.2. t-Distributed Stochastic Neighbor Embedding

After extracting features from raw data, one of the methods to evaluate the quality of the extracted features is data visualization. However, since the features generally have two or more characteristics, it is necessary to transform the features to visualize with two or three characteristics. One commonly used methods for this purpose is t-distributed stochas-

tic neighbor embedding (t-SNE) [22,23]; t-SNE is one of the nonlinear dimensionality reduction methods [24]. First, high-dimensional data is randomly expressed in low-dimensional data. Then, in the original feature space, points that are close to each other are placed close to each other, and points that are far from each other are placed farther away from each other. While continuously updating the position of the data in the low dimension, it aims to make the similarity between the data in the high dimension and the similarity between the data in the low dimension become similar [23,25]. Stochastic neighbor embedding (SNE) expresses the similarity with the conditional probabilities $p_{j|i}$ and $p_{i|j}$, instead of using the Euclidean distance to calculate distances in high and low dimensions. In Equations (1) and (2), suppose $x_i$ and $x_j$ are high-dimensional data, and $y_i$ and $y_j$ are the corresponding low-dimensional data, the similarity expressed as

$$p_{j|i} = \frac{exp\left(-\|x_i - x_j\|^2/2\sigma_i^2\right)}{\sum_{k \neq i} exp\left(-\|x_i - x_k\|^2/2\sigma_i^2\right)} \tag{1}$$

$$q_{j|i} = \frac{exp\left(-\|y_i - y_j\|^2/2\sigma_i^2\right)}{\sum_{k \neq i} exp\left(-\|y_i - y_k\|^2/2\sigma_i^2\right)} \tag{2}$$

where $\sigma_i$ is the standard deviation of the Gaussian distribution centered on $x_i$, which is determined by the data used for analysis. In SNE, the Kullback–Leighbler divergence between the two distributions is minimized to find the low-dimensional embedding $y_i$ that makes $p_{j|i}$ and $q_{j|i}$ similar to the given high-dimensional data $x_i$ [23]. Value $y_i$ is found using the gradient descent method. However, SNE uses asymmetric conditional probability and has a crowding problem in that it cannot embed data far from high dimensionality far enough away from low dimensionality. As an alternative to this, t-SNE uses the similarity $p_{ij}$ of a symmetric structure such as Equation (3) at a high level.

$$p_{ij} = \frac{p_{j|i} + p_{i|j}}{2} \tag{3}$$

Additionally, to solve the crowding problem, the student t-distribution with 1 degree of freedom is used as in Equation (4) to obtain the similarity $q_{ij}$ in the low dimension.

$$q_{ij} = \frac{\left(1 + \|y_i - y_j\|^2\right)^{-1}}{\sum_{k \neq l}\left(1 + \|y_i - y_l\|^2\right)^{-1}} \tag{4}$$

In the process of finding the low-dimensional embedding $y_i$, similar to SNE, the Kullback–Leighbler divergence of Equations (3) and (4) is minimized.

*2.3. Support Vector Machine for Multiclass Classification*

There are two main types of machine learning methods: supervised learning and unsupervised learning. Supervised learning is usually used for classification and regression problems, while unsupervised learning is used for clustering and dimension reduction. If there are labels for certain conditions of the system and we need to match the label of the new input condition, classification based on supervised learning can be used [22]. Supervised learning-based classification includes support vector machine (SVM) and artificial neural network (ANN) as representative examples. ANN consists of input layer, hidden layer, output layer, and neurons in each layer, and aims to obtain correct answer labels from input data by updating weights through feed forward and back propagation. ANN learns inherent features from given data by itself [25], so a large amount of high-quality data is needed for learning [26]. SVM aims to find the maximum marginal hyperplane (MMH) that maximizes the margin between each class [27,28]. It can also correspond to a nonlinear problem in the input space with a linear problem in a high-dimensional feature space, and relatively good prediction performance can be expected even with a small number of samples.

The process of calculating MMH with SVM is as follows [28]. Suppose that there are two types of classes. Given a training set $D = \{x_i, y_i\}_{i=1}^{N}$ for the binary classification problem, in which the class of $x_i \in \mathbb{R}^p$ is denoted by $y_i \in \{+1, -1\}$. Assume that a separation hyperplane, which means decision boundary, that separates data according to class is $\omega \cdot x + b = 0$. Here, $x$ is the point on the hyperplane, $\omega$ s the vector perpendicular to the hyperplane, and $b$ is the bias. When training data can be linearly separable, the hyperplane has the following properties:

$$\omega \cdot x_i + b \geq 1, \ y_i = 1$$
$$\omega \cdot x_i + b \leq -1, \ y_i = -1 \tag{5}$$

The training data located on the two hyperplanes described in Equation (5) is called a support vector (SV), and the distance between the two hyperplanes is called a margin. The margin can be expressed as $\frac{2}{\|\omega\|}$ and can be substituted with $\frac{1}{2}\|\omega\|^2$. Therefore, the objective function and constraint to find the MMH condition are the same as Equation (6). Equation (6) can be classified as a quadratic optimization problem of a convex function with constraints, and can be converted into a quadratic programming problem to calculate the boundary where the margin is maximal [29,30].

$$minimize \ \tfrac{1}{2}\|\omega\|^2$$
$$subject \ to \ y_i(x_i, \omega + b) \geq 1, \ i = 1, \ 2, \ \dots, \ N \tag{6}$$

If it is impossible to completely separate the training data linearly, soft margin classification can be performed by adding a slack variable $\xi_i$ to Equation (6) to impose a penalty $C$ for errors. In addition, since most class classification problems of data are complex, a non-linear support vector machine (non-linear SVM) is used. Non-linear SVM maps a feature vector $x_i$ in the original space to a new high dimensional space $\Phi(x_i)$ to calculate a nonlinear decision boundary. In this case, a kernel trick is used [30]. Instead of calculating the dot product after mapping the data to a higher dimension, the calculation cost is reduced by performing the dot product in the original dimension and sending it to the higher dimension. Popular kernel functions to construct the decision rules include the radial basis function (RBF) kernel, the polynomial kernel, and the multilayer perceptron (MLP) kernel. The objective function and its constraint to find the condition of MMH based on soft margin classification using kernel trick are the same as Equation (7).

$$minimize \ \tfrac{1}{2}\|\omega\|^2 + C \sum_{i=1}^{N} \xi_i$$
$$subject \ to \ y_i(\Phi(x_i), \omega + b) \geq 1, \ i = 1, \ 2, \ \dots, \ N \tag{7}$$

Originally, SVM was designed for the purpose of binary classification, but in the case of a multiclass classification problem, it can be solved by generalizing the SVM to a multiclass classifier or adopting a strategy that uses binary SVMs as multiple [28]. In the former case, the computational complexity is high and the performance is similar to the latter method, so it is preferred. In the latter case, there are one against all (OAA), one against one (OAO), and error-correcting output coding (ECOC) methods [31–33]. Among them, the OAO technique showing the best performance is widely used. The OAO technique learns a classifier using data belonging to two classes for each possible combination pair of classes. Afterwards, when new data comes in, each classifier classifies and votes to classify the data into the class with the most votes. Through this procedure, the SVM designed as a binary classifier can be used for multiclass classification.

### 2.4. Evaluation Metrics of Multiclass Classification Performance

The results of classifying data using machine learning methods are divided into four categories, which are true positives (*TP*), true negatives (*TN*), false positives (*FP*), and false negatives (*FN*). Suppose that a positive as an actual faulty state, and a negative is not the

faulty state, which contains a normal state. *TP* classifies the real fault as a the positive, and *TN* indicates the non-fault as a negative. On the other hand, *FP* and *FN* mean the misclassified case. When the faulty data is mistakenly labeled as the normal, it is called *FN*. Conversely, when the normal data is misclassified as the fault, it is called *FP*.

It is possible to evaluate the diagnosis performance of the machine learning model by calculating several evaluation metrics, listed as Equations (8)–(11), using the above-mentioned categories [2]. *Accuracy* refers to the proportion of data that accurately predicted among the total data. *Precision* represents the percentage of correct answers among the positive predictions. *Recall* indicates the percentage predicted by correct answers among actual faults. $F_1$ *score* is a score using the harmonic mean of *Precision* and *Recall*.

$$Accuracy = \frac{TP + TN}{TP + FP + FN + TN} \tag{8}$$

$$Precision = \frac{TP}{TP + FP} \tag{9}$$

$$Recall = \frac{TP}{TP + FN} \tag{10}$$

$$F_1 \ score = 2 \times \frac{Precision \times Recall}{Precision + Recall} \tag{11}$$

*Accuracy* is an effective metric when data is balanced, so it is used when the number of input data of each class is the same. However, when the data is not balanced, the $F_1$ *score* can be utilized. In multiple classification, $F_1$ *score* may be obtained as an average of $F_1$ *scores* of each class or may be obtained as an average of $F_1$ *scores*, considering the weight according to the data size of each class.

## 3. Establishment of Fault Diagnosis Framework Using Fault Simulation

The development of the CBM with PHM framework starts with establishing a condition database for the target system. After that, deriving features representing the system status through signal pre-processing, it proceeds in the order of developing a framework for fault diagnosis and prognosis [6]. In this section, we describe the data acquisition process and the fault diagnosis procedure for the development of a fault diagnosis framework based on condition monitoring of an oil purifier.

### 3.1. Data Acquisition through Fault Simulation

The first stage for the development of CBM with PHM is to acquire a lot of data with accurate information about the normal and the fault [6–8]. However, it is difficult to acquire the faulty data because of the costs and consequences of operating the system in a failure state [9]. In this study, we employed a strategy that simulates fault conditions and monitors the conditions by using the land-based test-bed of an oil purifier.

First, the target is Alfa Laval S805 shown in Figure 3. When the motor rotates at 3600 rpm, the bowl rotates at 9300 rpm by flat belt transmission and centrifugal force is generated. At the same time, the crude oil is separated into the purified oil and the sludge by density difference in the fluid inside the bowl. In order to acquire the normal and faulty state data from the test-bed, five general failure modes with the highest criticality were derived from failure modes, effects and criticality analysis (FMECA) in the previous research [3]. Table 2 describes test conditions with the failure modes to simulate the faults in the oil purifier. Criterion for simulating each failure mode refer to the standards provided by Alfa Laval [12], and failures are simulated under the most severe conditions within the range specified in the standards, or further severe condition than that.

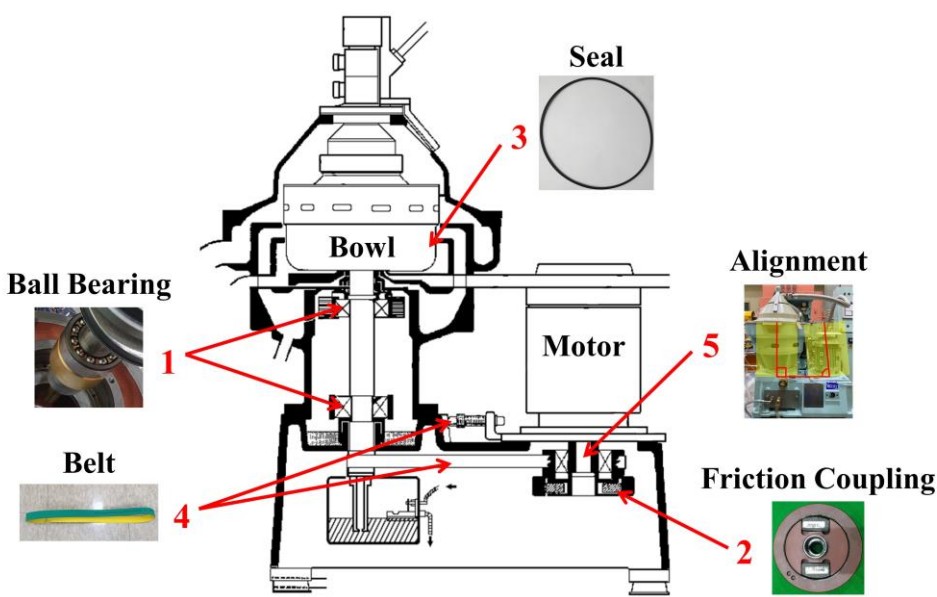

**Figure 3.** Schematic showing parts with high failure criticality of the oil purifier.

**Table 2.** Five types of failure components, modes, mechanisms, and effects.

| Mode Number | Component | Failure Mode | Failure Mechanism | Test Condition |
|---|---|---|---|---|
| 0 | - | Normal state | - | - |
| 1 | Bearing | Boundary lubrication | Poor lubrication | Reducing lubricant 100% compared to full capacity |
| 2 | Friction coupling | Poor power transmission | Wear of friction elements | Reducing the element mass 100% compared to the original state |
| 3 | Seal | Tension degradation | Thermal aging | Thermal aging at 120 °C for 900 h |
| 4 | Flat belt | Fatigue | Tension degradation | Shifting the motor towards the bowl by 13.62 mm |
| 5 | Shaft (=Spindle) | Fatigue Crack | Misalignment | Tilting the motor shaft by 0.9° counterclockwise |

In order to acquire the operating state through the fault simulation, the test-bed for condition monitoring was constructed as shown in Figure 4. Pressure, temperature, and flow data were acquired to monitor the operation. Because the change in the operating state does not occur instantaneously, the data on the operating state were sampled by 1 kHz, respectively. In addition, to understand additional failure response characteristics, vibration, noise, and RPM data were acquired at 25.6 kHz, 51.2 kHz and 1 kHz, respectively. The accelerometers to acquire vibration measured vibration of 3-axis and were located on the upper part of the bowl frame and the upper part of the motor frame. One acoustic emission sensor was used to acquire the noise data, and it was located at a point 1 m away from the test-bed. In addition, an RPM sensor was installed to check the power transmission of the motor by friction coupling. After applying the faults for each part, data on the normal and the fault condition was acquired from the built test-bed. The status of the test-bed was monitored for 30 min, and the acquired data was integrated and saved in 'csv' format.

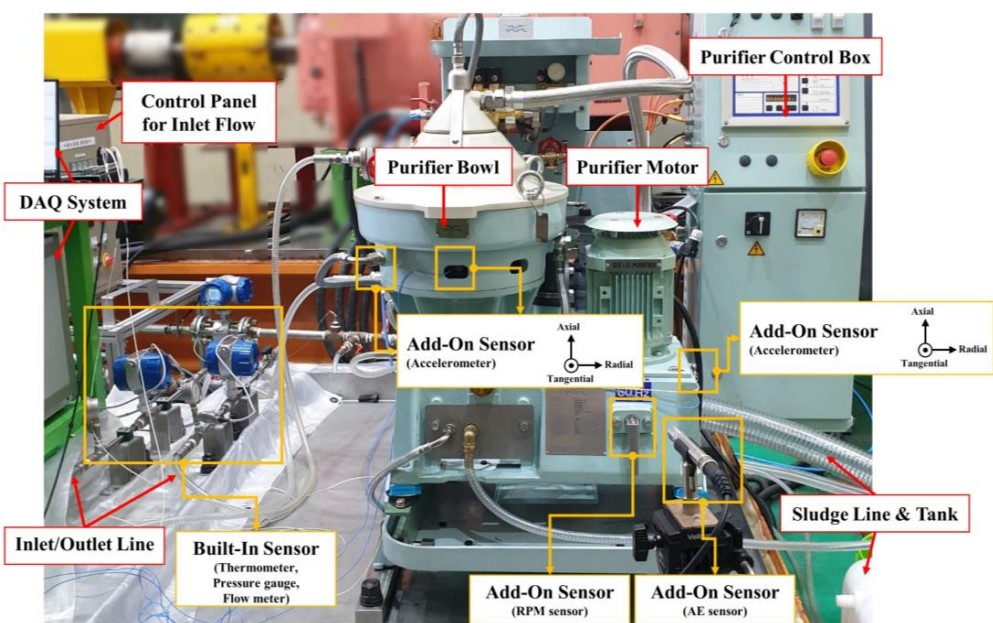

**Figure 4.** Oil purifier test-bed for land-based fault simulation.

As a result of monitoring the state of the test-bed for each failure mode, significant differences were observed according to the failure mode in the vibration data. Figure 5 shows the results of FFT of vibration in the axial and radial directions of the bowl, and Figure 6 shows the results of FFT of vibration in the axial and radial directions of the motor, respectively. Modes 1–5 are as shown in Table 2, and mode 0 indicates normal condition.

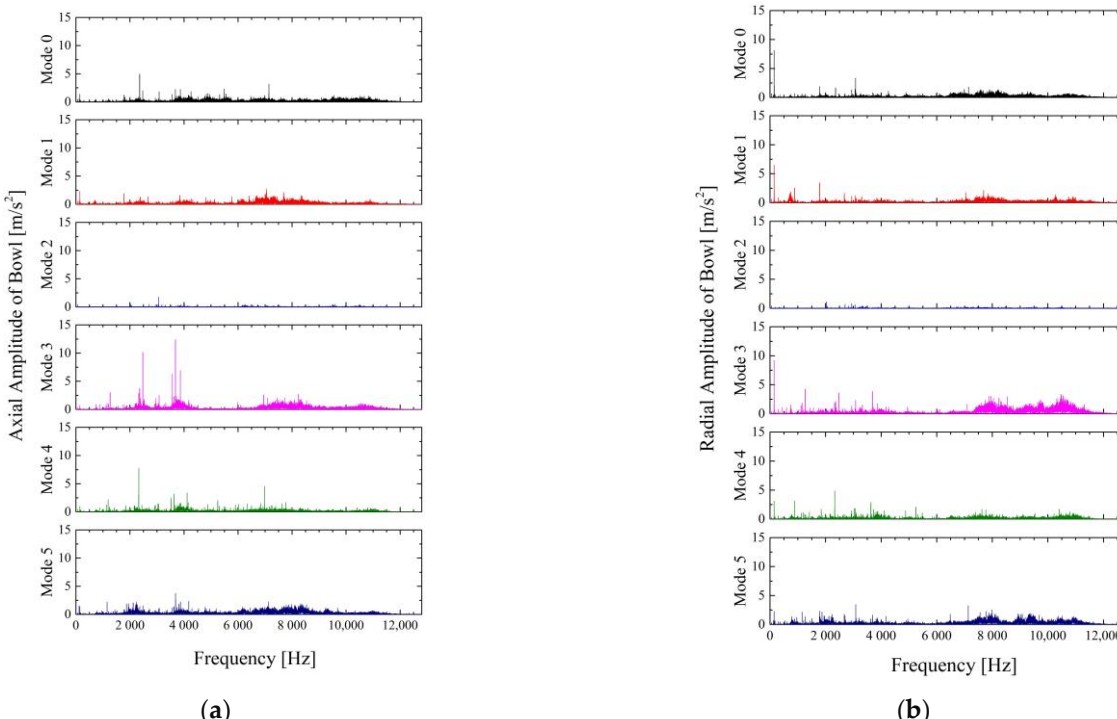

(**a**)                                                   (**b**)

**Figure 5.** The FFT graph of the vibration from the oil purifier bowl with respect to the measurement direction, which indicates mode 0 to 5 from the top. (**a**) Radial vibration of the bowl; (**b**) Axial vibration of the bowl.

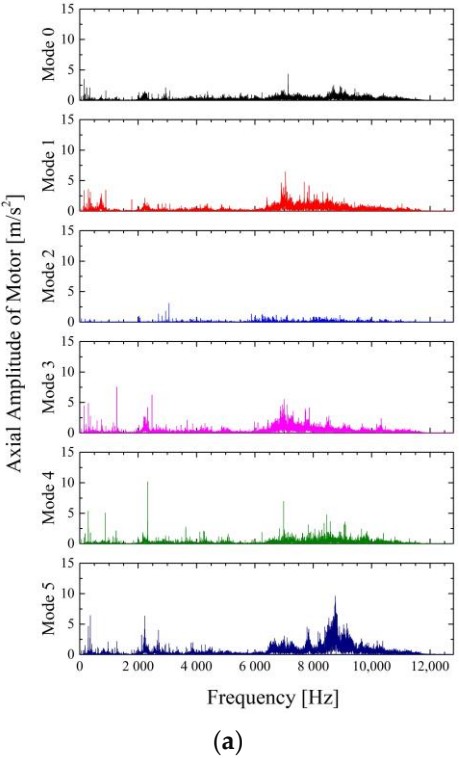
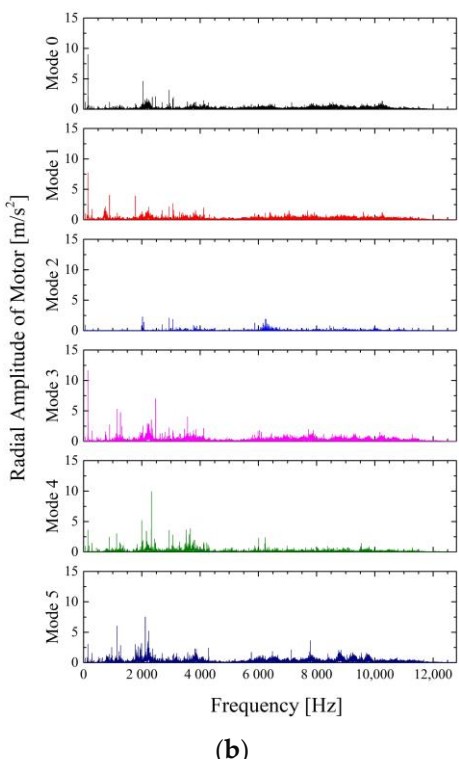

<div align="center">(<b>a</b>)　　　　　　　　　　　　　　　　　　　　　　　(<b>b</b>)</div>

**Figure 6.** The FFT graph of the vibration from the oil purifier motor with respect to the measurement direction, which indicates mode 0 to 5 from the top. (**a**) Radial vibration of the bowl; (**b**) Axial vibration of the bowl.

Overall, the motor vibration and bowl vibration are similar, or the motor vibration is slightly larger. This is because the place where the motor side sensor is attached vibrates relatively more than the place where the bowl side sensor is installed. In most modes except for mode 2, peaks can be seen at the motor frequency 60 Hz and the bowl frequency 155 Hz. Since mode 2 simulates the mass reduction due to friction of the friction coupling element, it means that the steady state RPM of the bowl could not be reached because the power could not be sufficiently transmitted in the defective state. In the case of mode 3, in Figure 6b, local gap and leakage occurred due to thermal degradation of the seal, resulting in mass imbalance. Therefore, it is judged that the bowl frequency amplitude has increased compared to the normal state. In case of other failure modes, it is difficult to confirm the significant amplification of the peak compared to mode 0 at motor frequency or Bowl frequency. This is the result because the standards suggested by the manufacturer are within the operating range of the equipment.

No significant difference was found in the time series data that monitored the operating status except for the mode 2. The faults were applied to each part, but such faults are considered not to change the operating state of the entire system. Additionally, ambient and sensor noise was observed in the pressure, temperature, and flow data. Therefore, in this study, the vibration data was selected as a monitoring factor and used for fault diagnosis. In addition, for each mode, the most prominent faulty data, which means the most severe faulty condition, were utilized.

### 3.2. Framework Organization for Fault Diagnosis

Since the vibration data selected as the monitoring factor was sampled at 25.6 kHz for 30 min, the size of the data was large, and it was not easy to analyze. Accordingly, a framework was constructed using feature extraction and machine learning-based diagnosis. First, in order to extract the features, the original signal was decomposed using discrete wavelet, and a few wavelet spectrum measures were calculated to extract the features.

Second, for multiclass diagnosis, a support vector machine-based multi-classification model was trained. Finally, we evaluated the performance of the diagnosis model using *Accuracy* and *F*₁ *score*.

### 3.2.1. Feature Extraction Using Discrete Wavelet Decomposition

When the oil purifier is running, it takes about 10 min for the bowl to reach the target RPM. Since the raw signal was sampled for 30 min from the starting point of the operation, it includes both steady state and non-steady state. Therefore, vibration data between 10 and 30 min after the motor rotation speed reached a steady state were used. The vibration data was sampled at 25.6 kHz, the truncated data was divided into 1 s-frame data that can be analyzed at frequencies up to 12.8 kHz. Then, WPT and DWT were performed with the divided data as input. The Daubechies function were used as a wavelet function. To determine the classification accuracy for degree of the decomposition, the signal was decomposed by changing the degree $j = 7, 8, 9$. At last, four wavelet spectrum measures in Table 1 were calculated.

The results of DWT $j = 8$ of the radial direction vibration of motor are shown in Figures 7 and 8 shows the WPT $j = 8$ results. Regardless of the signal preprocessing method, Failure mode 2 indicated clear differences, unlike the rest of the modes. In the case of wavelet spectrum using DWT, slight differences were observed among the failure modes except the mode 2, in Figure 7a,c. In Figure 7b,d, significant differences by the failure mode were shown in the 3rd and 7th nodes, which had a frequency band of 100–200 Hz and 1600–3200 Hz. In the case of wavelet spectrum using WPT, Figure 8b,d exhibit that there were significant differences near the 64th node as well as at the 3rd and 4th nodes. The 3rd and 4th nodes of WPT $j = 8$ had a frequency band of 100–150 Hz and 150–200 Hz, which contained the dominant bowl frequency. However, the rest of the nodes had similar values.

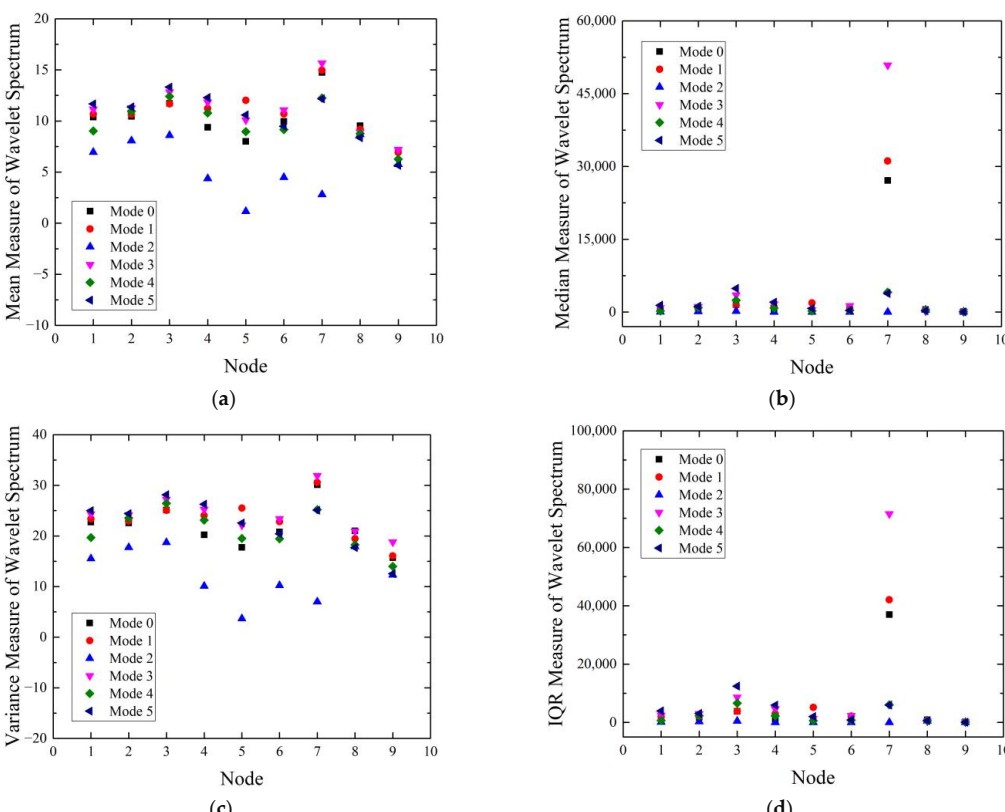

**Figure 7.** The wavelet spectrum measure of DWT $j = 8$ with respect to the failure mode in the radial direction of the motor. (**a**) mean; (**b**) median; (**c**) variance; (**d**) IQR.

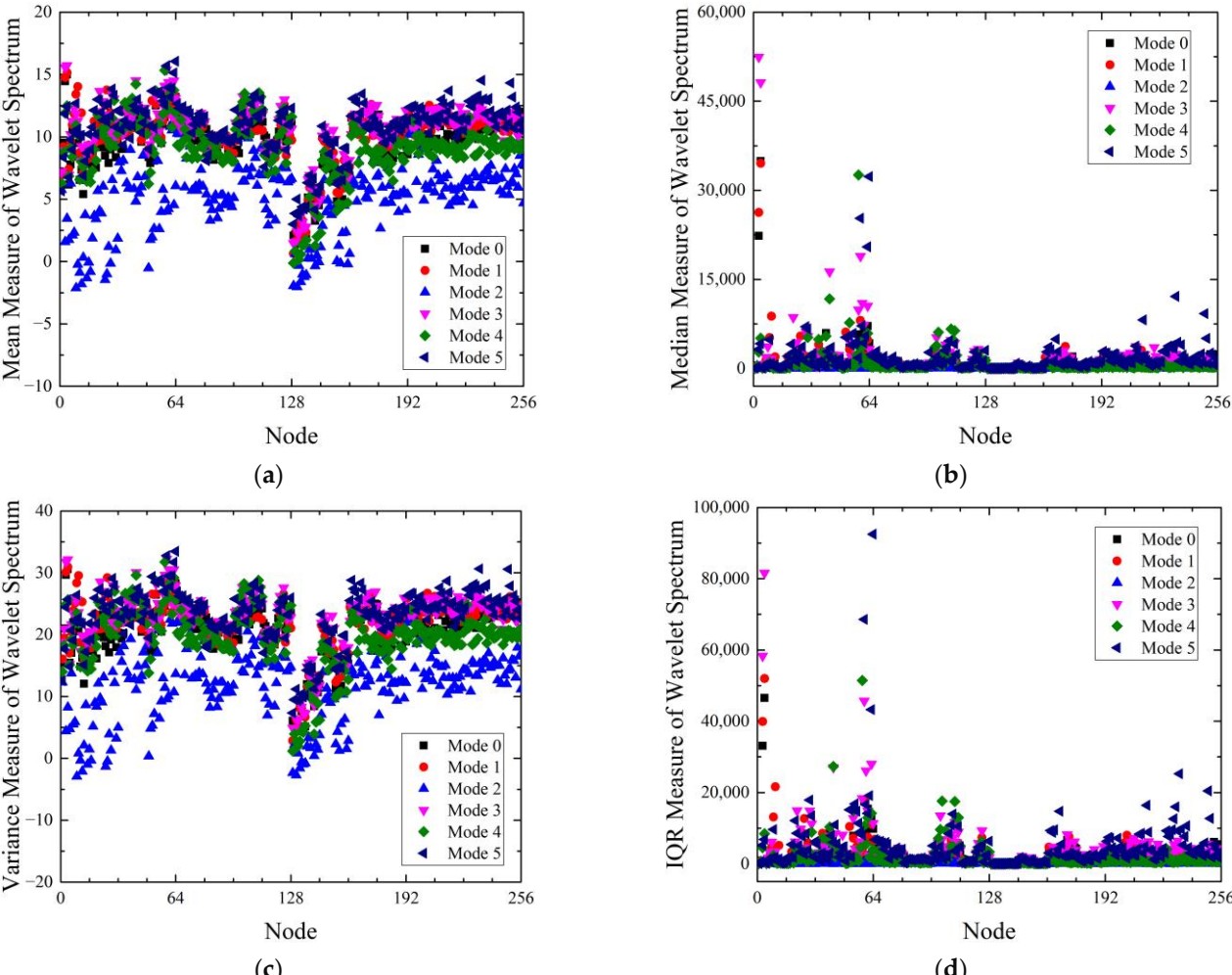

**Figure 8.** The wavelet spectrum measure of WPT *j* = 8 with respect to the failure mode in the radial direction of the motor. (**a**) mean; (**b**) median; (**c**) variance; (**d**) IQR.

In Figure 8b,d, the nodes with significant differences were the same as the part with the frequency band amplified vibration, as seen in Figures 5b and 6b. It was confirmed that the WPT can obtain a higher resolution and more frequency information than the DWT because it decomposes all frequency bands. However, except for a few nodes that showed significant differences according to the failure mode, most of them had similar values, so it is hard to evaluate that the feature extraction was successful.

### 3.2.2. Feature Visualization Using t-SNE

To evaluate the features extracted by the above process, t-SNE, one of the visualization methods, was used. The t-SNE results on four wavelet spectrum measures of the motor radial direction extracted through the DWT and the WPT are shown in Figures 9 and 10. In Figure 9, when using the DWT, it is exhibited that median and IQR measures distinguished each failure mode well. However, it is observed that a single cluster was not formed as the decomposition level increases. Most of them were classified well, but the black data with the mode 0, meaning the normal condition, did not form a single cluster and spread at decomposition level *j* = 7 and *j* = 8.

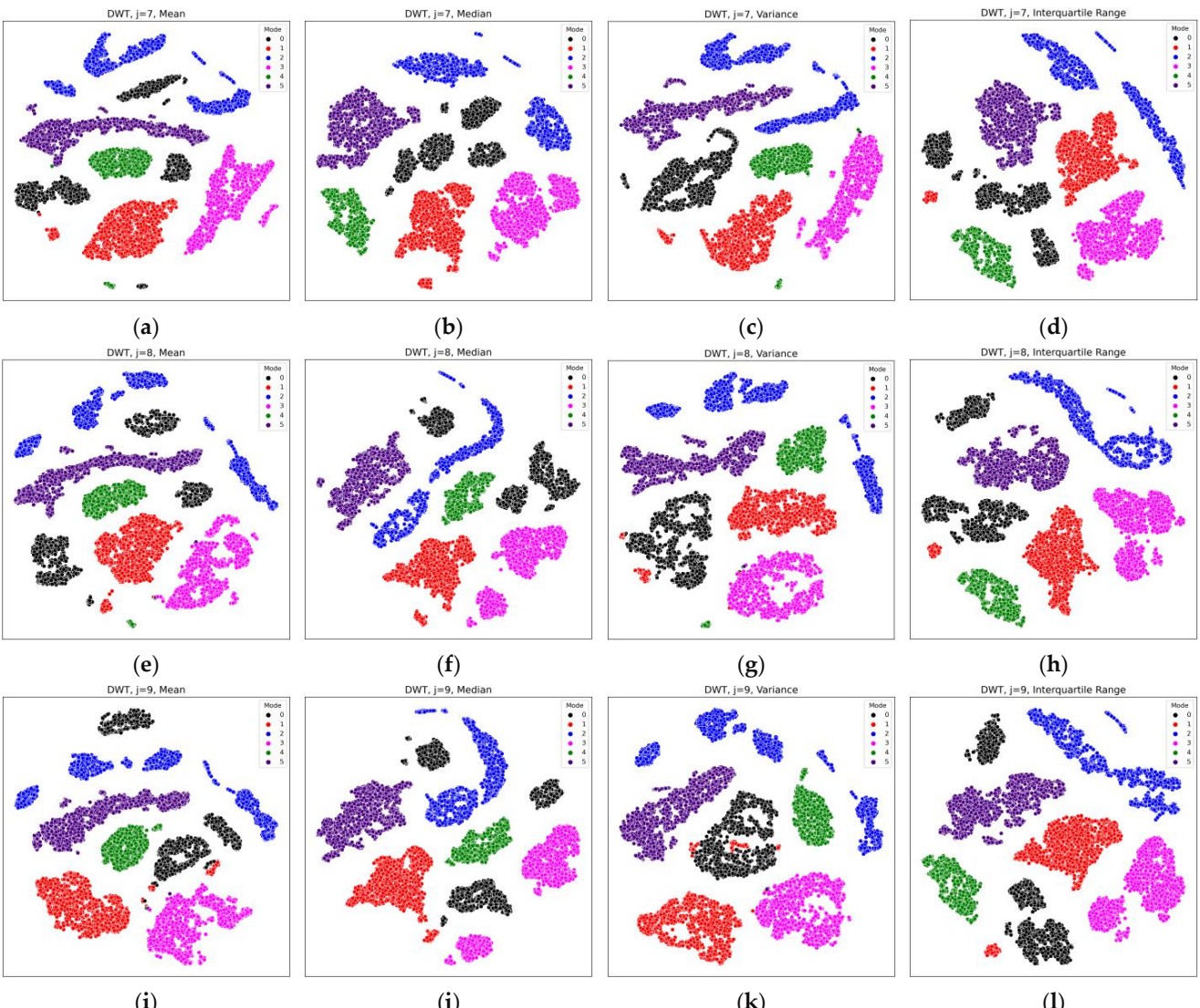

**Figure 9.** The t-SNE result of wavelet spectrum measures with respect to the decomposition level *j* based on the DWT. Wavelet spectrum measure varies in the row direction. (**a**,**e**,**i**) mean; (**b**,**f**,**j**) median; (**c**,**g**,**k**) variance; (**d**,**h**,**l**) IQR. The decomposition level *j* increases in the column direction. (**a**–**d**) *j* = 7; (**e**–**h**) *j* = 8; (**i**–**l**) *j* = 9. The failure modes 0 to 5 are represented in black, red, blue, magenta, green, and indigo colors, respectively.

When the WPT was used, it is confirmed that each mode formed a cluster well when using median and IQR measures in Figure 10. Similar to the t-SNE of DWT, it indicates that single cluster for each mode could not be formed as the decomposition level increased. In particular, the blue data with mode 2, which means wear of the friction element, was widely distributed here. However, when the WPT was used, clusters were locally formed, unlike the DWT. The results of t-SNE describes that the more the low frequency band was decomposed, the more difficult it was to form a single cluster, and the more the high frequency band was decomposed, the more each cluster was localized.

### 3.2.3. Fault Diagnosis Using OAO SVM

To take account of non-linearity included in the data, the non-linear SVM using the RBF kernel was employed. In addition, the OAO SVM implemented by using Python *scikit-learn* package was adopted to classify the six classes. The learning parameters of the OAO SVM were fixed as $\gamma = 0.1$ and $C = 10$, so that we could evaluate the diagnostic performance of the model with respect to the different input data.

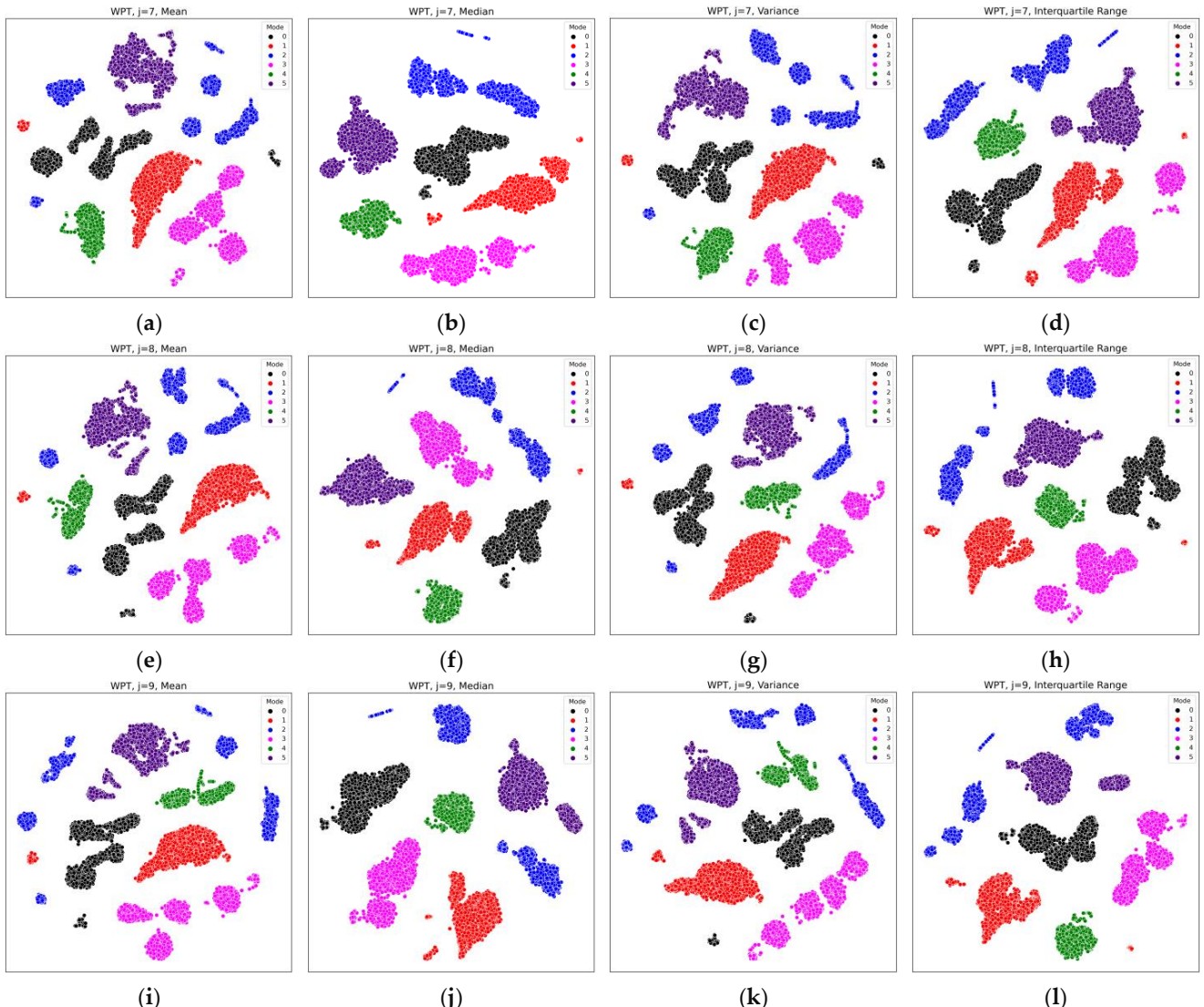

**Figure 10.** The t-SNE result of wavelet spectrum measures with respect to the decomposition level *j* based on the WPT. Wavelet spectrum measure varies in the row direction. (**a**,**e**,**i**) mean; (**b**,**f**,**j**) median; (**c**,**g**,**k**) variance; (**d**,**h**,**l**) IQR. The decomposition level *j* increases in the column direction. (**a**–**d**) *j* = 7; (**e**–**h**) *j* = 8; (**i**–**l**) *j* = 9. The failure modes 0 to 5 are represented in black, red, blue, magenta, green, and indigo colors, respectively.

In terms of input data, features that increase the diagnostic accuracy of the entire class and features that increase the diagnostic accuracy of a specific class may be different from each other. For this reason, the four wavelet spectrum measures of all nodes were used as input data, respectively, which mean a classification model was generated one by one for each type of wavelet decomposition and wavelet spectrum. The total data set consisted of 22,800 data, 13,200 data for training and 9600 data for testing. Then, in the training set, it was divided again according to a ratio of 8:2, so that 10,560 data were used for training, and 2640 data were used for validation. At first, the model was trained by setting the number of iterations to 100, and then the change in accuracy on the validation set according to iteration was observed.

In Figure 11, with increasing iteration, it was observed that the validation accuracy increased when each DWT-based wavelet spectrum measure of the radial vibration of the motor was used in training. Similarly, the validation accuracy of the model trained using that of the bowl data increased, as shown in Figure 12. When the radial vibration of the bowl was used, the validation accuracy improved more slowly than when the motor radial

vibration was used. However, when each WPT-based wavelet spectrum measure was used, it showed that the validation accuracy did not increase when *j* = 9 or decreased gradually when *j* = 7 and *j* = 8, in Figure 11b,d and Figure 12b,d. Additionally, it was observed that the validation accuracy of the mean and variance measures based-model was higher than that of the median and IQR measures-based model. In the case of the mean and variance measures, the training was relatively well performed because they had balanced slight differences in all nodes compared to the other measures-based models.

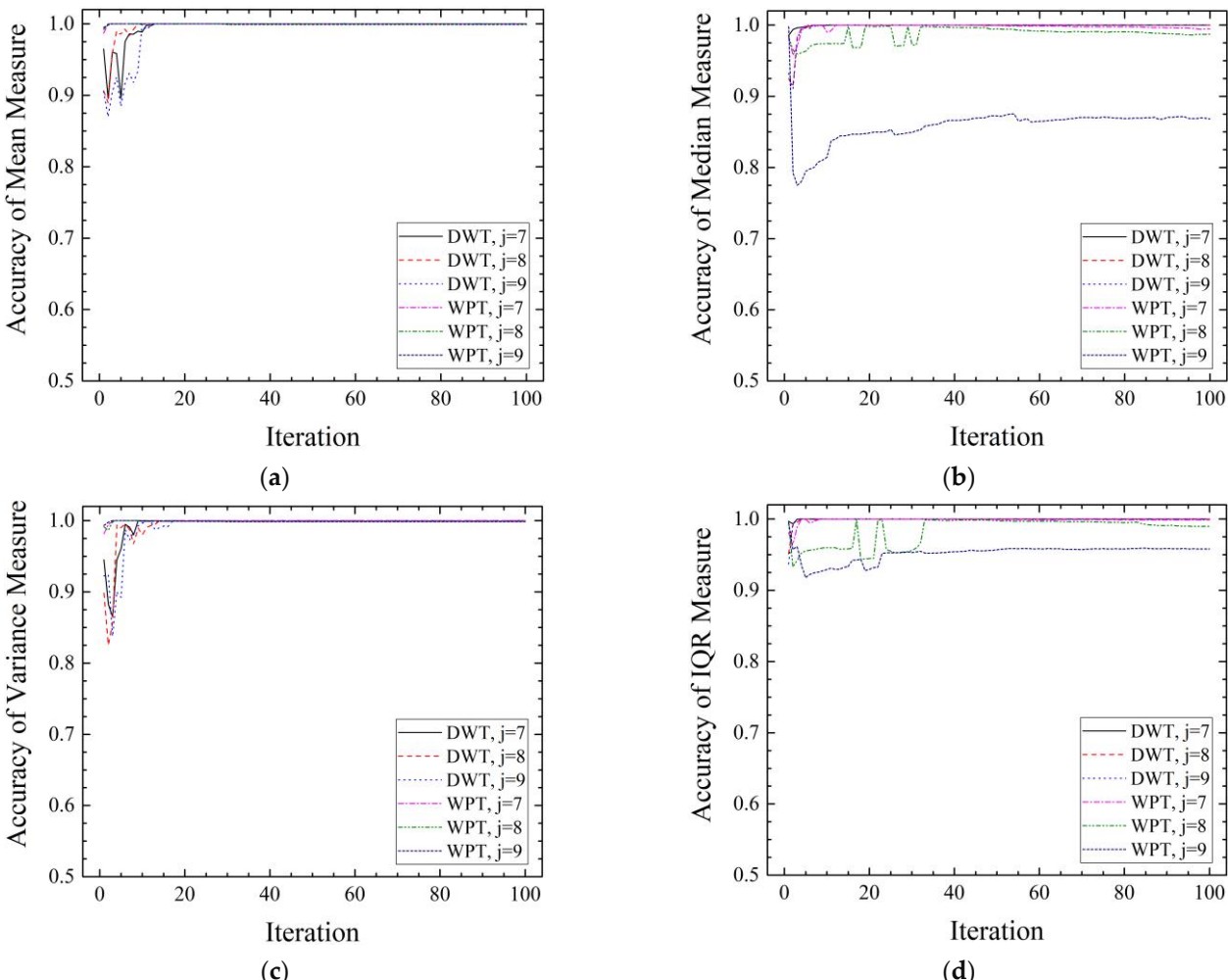

**Figure 11.** The validation accuracy of each wavelet spectrum measures of the motor radial vibration with increase in iterations with respect to the decomposition methods and decomposition levels. (**a**) mean; (**b**) median; (**c**) variance; (**d**) IQR.

In Figures 11 and 12, the validation accuracy for the radial vibration of the motor and the radial vibration of the bowl increased in different ways, but most of the models showed more than 90% of the diagnostic accuracy after 50 iterations. In terms of measurement position, these results suggest that both positions have useful features for fault diagnosis. In addition, when the WPT-based median and IQR measures were repeatedly trained 50 times or more, the result of the validation accuracy being the same or slightly lowered can be considered overfitting. It seems that as more signals were decomposed, the number of features irrelevant to classification increased, which reduced the explanatory power of the model at the same number of iterations.

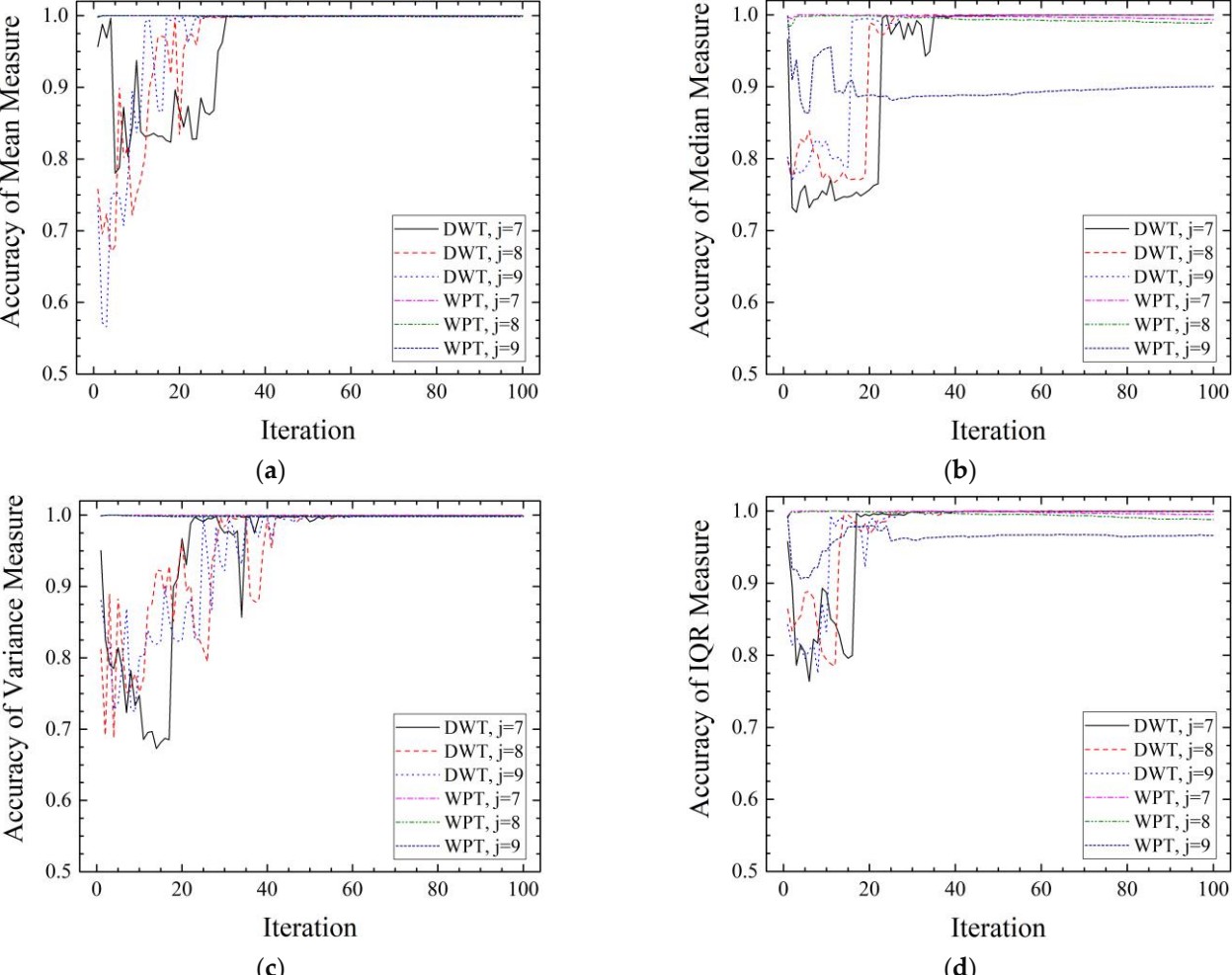

**Figure 12.** The validation accuracy of each wavelet spectrum measures of the bowl radial vibration with increase in iterations with respect to the decomposition methods and decomposition levels. (**a**) mean; (**b**) median; (**c**) variance; (**d**) IQR.

### 3.2.4. Evaluation of Diagnosis Performance

In the previous results, when repeatedly training the OAO SVM model with WPT-based data about 50 iterations, it was observed that the validation accuracy decreased rather than increased. Thus, the number of the iteration was set to 50. Afterwards, *Accuracy* and $F_1$ *score* were used to evaluate the diagnostic performance of the trained model.

Table 3 shows the diagnostic performance of the OAO SVM model trained 50 times. When training with the DWT-based wavelet spectrum measures, *Accuracy* and $F_1$ *score* were more than 0.99 when $j = 7$ and $j = 8$. However, when the raw data were decomposed 9 times, *Accuracy* and $F_1$ *score* decreased. Additionally, as the decomposition level j increased, it is exhibited that the score of the axial direction data was lower than the score of the radial direction data. When training with the WPT-based wavelet spectrum measures, it is remarkable that *Accuracy* and $F_1$ *score* scored 1.000 at $j = 7$. Additionally, they decreased sharply as the decomposition level increased, which can observe at WPT $j = 7$ and $j = 8$ in Table 3.

Considering the high diagnostic scores were obtained when the DWT-based decomposition was used, both the motor and the bowl are suitable positions for diagnosing the faults in the oil purifier test-bed. In the terms of the diagnostic performance, the decrease in the *Accuracy* and $F_1$ *score* as the decomposition level increased may suspect overfitting. In particular, both methods further commonly decompose the low frequency band as the decomposition level increases. This demonstrates that there were features correlated to

the faults in the low frequency band. Therefore, it is presumed that further decomposition of the low frequency signal produced additional non-correlated features with the faults and reduced the diagnostic performance. Additionally, we confirmed the *Accuracy* and $F_1$ *score* decreases rapidly when WPT is used. Considering these results collectively, it can be inferred that the high-frequency band signals were relatively uncorrelated with the faults. Thus, using the WPT caused greater overfitting than using the DWT, which can be explained in the same context as forming localized clusters when using the WPT in t-SNE. It seems that the overall data used for training clearly had significant differences, but most of them consisted of the information of the high frequency band, which led each model to have reduced explanatory power.

**Table 3.** The *Accuracy* and $F_1$ *score* for performance evaluation of the trained OAO SVM model.

| Decomposition Method | | Wavelet Spectrum Measure | Bowl Axial Dir. | | Bowl Radial Dir. | | Motor Axial Dir. | | Motor Radial Dir. | |
|---|---|---|---|---|---|---|---|---|---|---|
| | | | *Accuracy* | $F_1$ *Score* | *Accuracy* | $F_1$ *Score* | *Accuracy* | $F_1$ *Score* | *Accuracy* | $F_1$ *Score* |
| DWT | $j = 7$ | Mean | 0.9992 | 0.9992 | 0.9991 | 0.9991 | 0.9961 | 0.9961 | 0.9975 | 0.9975 |
| | | Median | 0.9979 | 0.9979 | 0.9997 | 0.9997 | 0.9929 | 0.9929 | 0.9950 | 0.9950 |
| | | Variance | 0.9993 | 0.9993 | 0.9994 | 0.9994 | 0.9998 | 0.9998 | 0.9990 | 0.9990 |
| | | IQR | 0.9973 | 0.9973 | 0.9996 | 0.9996 | 0.9999 | 0.9999 | 0.9954 | 0.9954 |
| | $j = 8$ | Mean | 0.9597 | 0.9573 | 0.9994 | 0.9994 | 0.9135 | 0.8967 | 0.9970 | 0.9970 |
| | | Median | 0.9983 | 0.9983 | 0.9999 | 0.9999 | 0.9878 | 0.9877 | 0.9947 | 0.9947 |
| | | Variance | 0.9979 | 0.9979 | 0.9994 | 0.9994 | 0.9083 | 0.8888 | 0.9984 | 0.9984 |
| | | IQR | 0.9976 | 0.9976 | 0.9995 | 0.9995 | 0.9988 | 0.9987 | 0.9953 | 0.9953 |
| | $j = 9$ | Mean | 0.8833 | 0.8742 | 0.9809 | 0.9812 | 0.8815 | 0.8391 | 0.9968 | 0.9968 |
| | | Median | 0.9975 | 0.9975 | 0.9997 | 0.9997 | 0.9896 | 0.9895 | 0.9947 | 0.9947 |
| | | Variance | 0.8820 | 0.8832 | 0.9961 | 0.9962 | 0.8961 | 0.8683 | 0.9980 | 0.9980 |
| | | IQR | 0.9989 | 0.9989 | 0.9994 | 0.9994 | 0.9977 | 0.9977 | 0.9954 | 0.9954 |
| WPT | $j = 7$ | Mean | 0.9658 | 0.9666 | 1.0000 | 1.0000 | 1.0000 | 1.0000 | 1.0000 | 1.0000 |
| | | Median | 0.9246 | 0.9296 | 0.9095 | 0.9039 | 0.9602 | 0.9599 | 0.9375 | 0.9387 |
| | | Variance | 0.8855 | 0.8865 | 1.0000 | 1.0000 | 0.9996 | 0.9996 | 0.9994 | 0.9994 |
| | | IQR | 0.9298 | 0.9342 | 0.8988 | 0.8891 | 0.9902 | 0.9903 | 0.9209 | 0.9167 |
| | $j = 8$ | Mean | 0.8610 | 0.8592 | 0.8748 | 0.8748 | 0.8761 | 0.8772 | 0.8764 | 0.8767 |
| | | Median | 0.8390 | 0.8511 | 0.8609 | 0.8424 | 0.9305 | 0.9341 | 0.8532 | 0.8451 |
| | | Variance | 0.8590 | 0.8574 | 0.8744 | 0.8746 | 0.9531 | 0.9547 | 0.8747 | 0.8773 |
| | | IQR | 0.7788 | 0.7927 | 0.8488 | 0.8186 | 0.9425 | 0.9459 | 0.7740 | 0.7647 |
| | $j = 9$ | Mean | 0.7495 | 0.7141 | 0.7833 | 0.7565 | 0.6547 | 0.6345 | 0.6890 | 0.6870 |
| | | Median | 0.8186 | 0.8371 | 0.7131 | 0.6765 | 0.8073 | 0.8173 | 0.6992 | 0.7079 |
| | | Variance | 0.7698 | 0.7363 | 0.8132 | 0.8006 | 0.7595 | 0.7553 | 0.7542 | 0.7500 |
| | | IQR | 0.8447 | 0.8614 | 0.7485 | 0.7280 | 0.8103 | 0.8174 | 0.7267 | 0.7317 |

## 4. Conclusions

As the 4th industrial revolution is in progress, MASS has been emerging in the shipbuilding and shipping industry. For the development of Degree 4 MASS, the self-diagnosis technique is essential, and CBM with PHM for ship engine system facilities must be developed. In this article, we proposed a fault diagnosis framework using land-based fault simulation for the development of the CBM with PHM of an oil purifier, which is an auxiliary device for purifying oil in ships. In the first stage, we established the condition database of an Alfa Laval S805 oil purifier. The land-based test-bed of the oil purifier for fault simulation was built, and the data about the operating state were acquired by simulating the faults. In the second stage, the framework for fault diagnosis was organized, which included the processes of the feature extraction and fault diagnosis. First, raw signals were preprocessed by using the DWT and the WPT, which are decomposition methods enabling time-frequency analysis, and the features were extracted by calculating the wavelet spectrum. Next, the OAO SVM model was trained using the extracted features, and the diagnostic performance was evaluated with *Accuracy* and $F_1$ *score*.

In this study, we established the condition database by simulating the highly critical faults in the oil purifier, which can be used for developing CBM system of oil purifiers in real ships. In addition, using the proposed framework, it was confirmed that a high accuracy diagnosis of 0.99 or more is possible by utilizing just one accelerometer and measurement direction. The DWT and the WPT were performed for extracting features within the framework, and when features were extracted by using the WPT, better resolution was

provided compared to using the DWT. In Figures 7–10, it was observed that median and IQR measures rather than mean and variance measures could confirm a significant difference according to the failure mode.

However, when the OAO SVM model was trained with the more decomposed features, we could confirm the diagnostic performance decreased. Additionally, these were observed with both methods. At this point, we can suspect the overfitting. When the decomposition level was increased, detailed information could be obtained. However, since most of the data were composed of the high frequency features that were not correlated with the faults and they were all used for training, it seems that the explanatory power for the test data decreased. In this case, it is possible to increase the explanatory power of the model by selecting features or using low pass filters or band pass filters.

This study provides a reference for acquiring data using the test-bed in establishing a condition monitoring-based fault diagnosis framework for oil purifiers. However, the problem of lack of data still remains. After solving the problem and building a high-quality database, research should be conducted to approach fault diagnosis/prognosis based on data. One of the strategies for this is to utilize the test-bed adopted here, and another is to augment the necessary data quickly and efficiently by using generative models. Thus, based on the data obtained from the test-bed, we will conduct research on establishing a fault diagnosis/prognosis framework using the augmented data acquired by the generative models.

Additionally, the framework proposed in here will be verified with data acquired from the real ship soon. Sufficiently verifying and updating, it can be used for diagnosing faults in oil purifiers in real ships. After that, we will conduct research on prediction of a fault for a specific failure mode of oil purifiers using the test-bed. Through a series of research, the safety and reliability of oil purifiers in real ships can be secured, and it can contribute to the development of Degree 4 MASS.

**Author Contributions:** Conceptualization, T.L. and Y.K.; methodology, S.L. and T.L.; software, S.L.; validation, J.L. and K.R.; experiments, J.K., S.L. and J.L.; investigation, J.K. and J.L.; resources, T.L. and Y.K.; writing—original draft preparation, S.L.; writing—review and editing, T.L.; supervision, J.-W.P.; project administration, Y.K.; funding acquisition, T.L. All authors have read and agreed to the published version of the manuscript.

**Funding:** This work was supported by the Industrial Technology Innovation Program funded by the Minister of Trade, Industry & Energy (MOTIE) of the Republic of Korea (No. 20011164).

**Conflicts of Interest:** The authors declare no conflict of interest.

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
