# Peer review of "A Study on the Application of Discrete Wavelet Decomposition for Fault Diagnosis on a Ship Oil Purifier"

_processes, doi:10.3390/pr10081468_

Round 1
Reviewer 1 Report
1. In this paper, fault diagnosis framework was proposed for condition-based maintenance (CBM) of ship oil purifiers which are an auxiliary facility in engine system of a ship. The logic of this article is clear, but innovation seems insufficient. It is difficult to find out how the author has improved the actual existing methods.
2. In 3.2.1, why the truncated data can be analyzed at frequency up to 12.8kHz?
3. The proposed diagnostic method is not convincing. The vibration data at the time of failure should be collected again for method verification.
4. The proposed method is based on the vibration of the equipment. If considering the actual situation that the equipment is installed on the MASS, is the method still effective?
5. There are many pictures in this paper, but they lack comparison, so it is difficult to find any difference between adjacent pictures.
Reviewer 2 Report
This is a very pragmatic paper in which the authors propose a data-driven approach for fault diagnosis of purifiers by crete wavelet transform (DWT)
and wavelet packet transform (WPT), t-distributed stochastic neighbor embedding (t-SNE), and svm classification to improve the correct fault classification rate.
However, there are some flaws in the article. In the second section Methodology for Fault Diagnosis, the authors repeatedly mention many concepts and analyze and compare them, such as PCA. These concepts are not part of the model, but just a comparison of model methods. I suggest moving these to the introduction or related work.
Reviewer 3 Report
Comments are given below:
It's interesting, but it needs a lot of corrections.
1. After the purposes of the manuscript in the last paragraph of the introduction, the following explanations are not needed and should be deleted.
The remainder of this paper is organized as follows. In section 2, methodologies about feature extraction and machine learning for fault diagnosis are introduced. Section 3 describes the entire process for the development of a fault diagnosis framework. Finally, in section 4, the results are discussed, and conclusions are drawn.
2. In Table 2, only the misalignment failure mechanism is mentioned, there are other failures such as torsional and bending stress, why are the above items not mentioned.
3. Frequencies of 1, 25.6, and 51.2 kHz are selected for additional failure response, vibration, noise, and RPM data. It is selected from a specific reference or based on experimental data.
4. The graphs of figures 5 and 6 and also the graphs of figures 11 and 12 are not clear, so they need to be corrected.
5. The description of the quality images in Figures 9 and 10 is not clear in the text, it needs more explanation.
6. Due to the importance of the simulation method in the case study, the following refs may help the simulation method for verification and validation.
a) R. Masoudi Nejad, P. Noroozian Rizi, M.S. Zoei, K. Aliakbari, H. Ghasemi, Failure Analysis of a Working Roll Under the Influence of the Stress Field Due to Hot Rolling Process, J. Fail. Anal. Prev. (2021). https://doi.org/10.1007/s11668-021-01131-9.
b) K. Aliakbari, R. Masoudi Nejad, T. Akbarpour Mamaghani, P. Pouryamout, H. Rahimi Asiabaraki, Failure analysis of ductile iron crankshaft in compact pickup truck diesel engine, Structures. 36 (2022) 482–492.
Reviewer 4 Report
1. I consider that the topic is of actuality and scientifically challenging. The manuscript is clearly well structured and organized, it is easy to follow and the terminology is appropriate to the subject. Tables and figures are used efficiently and support the text, also the reference citations are complete and accurate
2. The content of the paper is succinctly described and contextualized in relation to the presented theoretical background.
3. The introduction provides a sufficient background on self-diagnosis techniques for ship engine design, in the context in which the concept of Maritime Autonomous Surface Ships (MASS) is currently promoted in the shipbuilding and shipping industry. Among these techniques the prognostics and health management (PHM) monitors complex system in real time, detects abnormalities in the system in advance, and predicts future failures beforehand.
I appreciate that in the introduction the authors present the problem to be solved and highlight their contribution. The authors clearly identify the area of interest, the issue to be addressed and its significance, and the purpose of the study is clearly expressed.
However, I recommend that you emphasize in a few sentences how your approach differs from other similar papers in the main flow of publications.
I recommend that you make sure that each statement made in the introduction will be demonstrated in the papers sections and refer the statements to their demonstration.
4. I consider that the cited references are relevant for research. I recommend that the authors highlight other papers that address the issue of diagnosing defects in shipbuilding and shipping.
5. The research design is appropriate. The authors propose a fault diagnosis framework for Condition Based Maintenance (CBM) of ship oil purifiers, which are auxiliary facility in engine system of a ship, CBM is complemented by prognostics and health management (PHM).
6. The concluding elements of the paper are represented by strong statements based on scientific arguments that are presented clearly and concisely. However, I believe that the authors should reflect the extent to which the results answered the questions mentioned in the introductory part: What is the research gab and what is your paper's contribution/ innovation for the research? In my opinion, the solid argumentation on the conclusions of the paper will open new research directions and lead to the deepening of the issues studied by potential readers.
7. We have found that bibliographic references (in total of 36) are described accurately, sincerely and deontologically by the authors.
8. I recommend the authors to present in a more promising matter the future research opportunities which are considered to be feasible and scientifically fertile in the field.
Round 2
Reviewer 1 Report
My questions have been answered. This paper can be accepted.
Reviewer 3 Report
Comments are given below:
Due to the corrections made to the revised manuscript in accordance with the comments requested, the manuscript is accepted as is.